# Transcriptomic Insights and the Development of Microsatellite Markers to Assess Genetic Diversity in the Broodstock Management of *Litopenaeus stylirostris*

**DOI:** 10.3390/ani14111685

**Published:** 2024-06-05

**Authors:** Ya-Chi Yang, Pei-Yun Chu, Che-Chun Chen, Wen-Chin Yang, Te-Hua Hsu, Hong-Yi Gong, I Chiu Liao, Chang-Wen Huang

**Affiliations:** 1Department of Aquaculture, National Taiwan Ocean University, 2 Beining Road, Jhongjheng District, Keelung City 20224, Taiwan; vicky877102@gmail.com (Y.-C.Y.); kimi0396@gmail.com (P.-Y.C.); fassadze@gmail.com (C.-C.C.); realgigi@gmail.com (T.-H.H.); hygong@mail.ntou.edu.tw (H.-Y.G.); 2Agricultural Biotechnology Research Center, Academia Sinica, No. 128, Academia Sinica Road, Sec. 2, Nankang, Taipei 11529, Taiwan; wcyang2024@gmail.com; 3Center of Excellence for the Oceans, National Taiwan Ocean University, 2 Beining Road, Jhongjheng District, Keelung City 20224, Taiwan; icliao@mail.ntou.edu.tw

**Keywords:** DNA molecular marker, genetic management, *Litopenaeus stylirostris*, next-generation sequencing, transcriptome

## Abstract

**Simple Summary:**

The Pacific blue shrimp (*Litopenaeus stylirostris*) is essential to the global seafood market, but intensive farming has increased disease and reduced genetic diversity. This study aimed to develop microsatellite markers for broodstock management using high-throughput transcriptome sequencing from various tissues. We identified 40 robust markers, revealing significant genetic diversity crucial for sustainable shrimp farming. These markers are valuable for breeding programs and help in the conservation and improvement of shrimp populations.

**Abstract:**

The Pacific blue shrimp (*Litopenaeus stylirostris*) is a premium product in the international seafood market. However, intensified farming has increased disease incidence and reduced genetic diversity. In this study, we developed a transcriptome database for *L. stylirostris* and mined microsatellite markers to analyze their genetic diversity. Using the Illumina HiSeq 4000 platform, we identified 53,263 unigenes from muscle, hepatopancreas, the intestine, and lymphoid tissues. Microsatellite analysis identified 36,415 markers from 18,657 unigenes, predominantly dinucleotide repeats. Functional annotation highlighted key disease resistance pathways and enriched categories. The screening and PCR testing of 42 transcriptome-based and 58 literature-based markers identified 40 with successful amplification. The genotyping of 200 broodstock samples revealed that *N*a, *H*_o_, *H*_e_, PIC, and *F*_IS_ values were 3, 0.54 ± 0.05, 0.43 ± 0.09, 0.41 ± 0.22, and 0.17 ± 0.27, respectively, indicating moderate genetic variability and significant inbreeding. Four universal microsatellite markers (CL1472.Contig13, CL517.Contig2, Unigene5692, and Unigene7147) were identified for precise diversity analysis in Pacific blue, Pacific white (*Litopenaeus vannamei*), and black tiger shrimps (*Penaeus monodon*). The transcriptome database supports the development of markers and functional gene analysis for selective breeding programs. Our findings underscore the need for an appropriate genetic management system to mitigate inbreeding depression, reduce disease susceptibility, and preserve genetic diversity in farmed shrimp populations.

## 1. Introduction

The sustainability of aquaculture has long been challenged by human factors, such as genetic breeding, feed nutrition, disease, water quality, and climate change [1]. The high fertility and early mortality of marine crustaceans can affect the stability of alleles in the population, thereby altering the genetic polymorphisms and structure of the natural population [2]. Over numerous breeding generations, populations exhibiting robust vigor and significant traits give rise to distinct genotypes, leading to a reduction in evolutionary plasticity [3] or the loss of genetic polymorphism caused by mutations or genetic drift [4]. The Convention on Biological Diversity has been closely focusing on the conservation of genetic diversity in both wild and farmed species, along with the development of effective methods for measuring it [5]. Therefore, the sustainable development of aquaculture also requires preserving genetic diversity in natural populations and maintaining genetic variability in farmed populations [6,7].

The Pacific blue shrimp (*Litopenaeus stylirostris*, Penaeidae) [8], also known as blue shrimp or western blue shrimp, is an economically valuable shrimp species with breeding potential [9]. It has gained popularity in foreign markets owing to its attractive flavor and high nutritional value [10,11]. Furthermore, *L. stylirostris* offers the advantages of rapid growth, disease resistance, wide salinity tolerance, high-temperature tolerance, low dissolved oxygen tolerance, and low food protein requirements. *L. stylirostris* are mainly distributed along the Pacific coasts of Latin America and Mexico, and their biological characteristics are similar to those of *L.vannamei*.

Pacific blue shrimp aquaculture not only meets the needs of consumers through shrimp farming but also reduces the potential fishing pressure on wild populations through stock enhancement [12,13]. However, environmental pollution and overharvesting have gradually reduced the number of natural blue shrimp populations [13]. This is evident from the increase in artificially bred shrimp populations, as demonstrated by the breeding data of shrimp seedlings in recent years. However, there is no systematic genetic management of breeding shrimps, and there is a lack of selection and breeding based on the body size and environmental adaptability of shrimps. This leads to varying body size and increased susceptibility to various diseases [14], such as yellow head disease [15], black gill disease, soft shell syndrome, and white muscle disease, which gradually develop during growth. Furthermore, the phenotypic depression of heritable traits in inbred species has led to a decline in the economic traits and production efficiency, resulting in serious economic losses to the shrimp industry [12,13]. Whether these problems arise from gene erosion in germplasms, or a lack of genetic management remains unclear [16,17]. Therefore, identifying the molecular markers to explore the genetic diversity and the strategic development of the genetic breeding management of this species is important [18].

The development of sequencing technologies, particularly next-generation sequencing (NGS), has revolutionized genomic research by generating vast amounts of transcriptome data across various species [19,20]. This advancement has facilitated the rapid identification of genetic markers, including microsatellites and genomic variations [21,22]. Importantly, NGS has enabled the annotation of functional genes and their associated traits, deepening our understanding of population genetics and evolutionary biology [23,24]. Transcriptome sequencing not only provides valuable insights into gene expression profiles but also allows for the annotation of functional genes, shedding light on the molecular mechanisms underlying various biological processes, including disease resistance in shrimp populations. Furthermore, the identification of microsatellites from transcriptome data offers a powerful tool for genetic analysis and management in aquaculture. These markers can be utilized to assess genetic diversity, population structure, and relatedness among individuals, aiding in the development of effective breeding strategies and conservation efforts [25,26].

Microsatellite markers are simple sequence repeats of highly polymorphic and codominant DNA motifs (n could range from 8 to 50) that are widely distributed throughout the genomes of eukaryotes and are widely used for different population genetic studies in many different species, such as genetic diversity [27,28,29], population structure, parentage determination [30,31,32], and constructing genetic linkage maps [33,34,35,36,37], animal identification, meat traceability [38,39] and animal breeding [40,41]. Studies on related aquatic species with highly duplicated genomes have demonstrated that microsatellites derived from transcriptomes are significantly more effective than those developed from the genome using traditional methods [42]. However, genetic studies on microsatellite markers and their related functions derived from next-generation transcriptome sequencing are limited. Furthermore, a few studies on other members of Palaemonidae, including *Macrobrachium nipponense* [23], *Macrobrachium rosenbergii* [43], and *Palaemonetes sinensis* [34], have explored the applicability of microsatellite markers in understanding the genetic diversity in these species. However, to date, no studies have been conducted on the genetic background of *L. stylirostris*-farmed populations using molecular markers related to functional genes in the transcriptome. We hypothesized that developing transcriptome-based microsatellite markers can help determine the possible reasons for the reduction in genetic diversity in the germplasm or gene weakening in *L. stylirostris*.

Therefore, the present study aimed to investigate the variation in different genotypes of transcriptome-derived microsatellite markers and analyze them in different cultured Pacific blue shrimp populations (different genetic backgrounds) using molecular marker-assisted selection to (i) rapidly identify whether the germplasm has gene-weakening problems and (ii) verify the applicability of the markers.

## 2. Materials and Methods

### 2.1. Experimental Animals

Pacific blue shrimps were collected from an experimental shrimp hatchery at the Golden Broodstock Development Center of the Golden Corporation Sendirian Berhad, Kampung Serambangun, Brunei, in July 2019. A total of 200 samples were obtained, comprising 100 male and 100 female shrimps from the broodstock population of the Pacific blue shrimp. This comprehensive sampling approach ensured a balanced representation of both sexes, allowing for a robust analysis of genetic diversity within the population. Detailed information on the size and growth stages of the shrimp was recorded. The average body weight was approximately 10.56 ± 0.26 g, and the body length ranged from 8 to 10 cm, categorizing them in the subadult stage. The subadult stage was selected because the shrimp were in a period of rapid growth, with high nutritional needs and stringent requirements for water quality and feed. This stage provides data that more accurately reflect current aquaculture conditions, offering valuable insights for future research. 

In light of the challenges involved in acquiring wild Pacific blue shrimp specimens, we followed the methods outlined [44] and opted to utilize commercially available frozen Pacific white shrimp (*L. vannamei*) and black tiger shrimp (*P. monodon*) as control samples. Thirty samples of commercially available fresh aquatic products of Pacific white shrimp were purchased from the Taiwan supermarket in December 2021. The samples were obtained from Fuguo Freezing Co., Ltd. in Yilan, Taiwan (produced in Taiwan), Charoen Pokphand Enterprise (Taiwan) Co., Ltd. in Taipei, Taiwan (produced in Thailand), and Lequality Life Co., Ltd. in Kaohsiung, Taiwan (produced in Nicaragua). The samples were labeled as PWS-TW (*n* = 10), PWS-TH (*n* = 10), and PWS-NI (*n* = 10), respectively. Thirty samples of commercially available fresh aquatic products of black tiger shrimp were purchased from the Taiwan supermarket in December 2021. The samples were obtained from Sanheng International Trading Co., Ltd. in New Taipei, Taiwan (produced in Malaysia), Shangyi Fishery Co., Ltd. in Taichung, Taiwan (produced in Malaysia), and First Flight Frozen Food Co., Ltd. in Kaohsiung, Taiwan (produced in Vietnam). The samples were labeled as BTS-MY1 (*n* = 10), BTS-MY2 (*n* = 10), and BTS-VN (*n* = 10), respectively (Figure A1). Tissue samples were stored at −80 °C until further RNA extractions were made.

To ensure that commercially available shrimp samples met the experimental requirements, several key evaluation steps were undertaken. First, the sex ratio was examined to ensure representativeness, involving visual inspections and random sampling of 10% of each batch to achieve an approximately 1:1 male-to-female ratio. Secondly, detailed data on body size and growth stages were recorded by measuring length and weight to determine appropriate developmental stages for the experiment. Third, health assessments included both external and internal inspections for signs of damage, disease, or parasites to ensure reliable results. Fourth, genetic diversity was assessed using known microsatellite markers to ensure sufficient genetic variation. Sample size and quality control measures, including repeated testing, were verified to ensure data reliability and consistency. Finally, the experimental design included control groups from different sources and growth stages to ensure result comparability, along with multiple replicates to validate the findings. Through these comprehensive evaluations, we ensured that the commercially available Pacific white shrimp and black tiger shrimp samples met all experimental requirements, guaranteeing reliable and valid research outcomes.

Each type of shrimp and tissue sample served specific purposes in our study: Pacific blue shrimp (*L. stylirostris*) samples were collected to establish a comprehensive genetic repository and analyze the genetic diversity within the broodstock population. Various tissues, including the muscle (MU), hepatopancreas (HE), lymphoid organ (LY), and intestinal tract (IN), were collected to capture a wide range of genetic information. Pacific white shrimp (*L. vannamei*) control samples were used as a genetic control to compare and validate the genetic markers identified in Pacific blue shrimp. These samples were obtained from different regions and labeled as follows: Taiwan (PWS-TW), Thailand (PWS-TH), and Nicaragua (PWS-NI). Black Tiger Shrimp (*P. monodon*) control samples provided additional genetic control and facilitated cross-species comparisons of the identified genetic markers. Fresh aquatic product samples were sourced from commercial suppliers. Samples were sourced from Malaysia (BTS-MY1), Malaysia (BTS-MY2), and Vietnam (BTS-VN), ensuring a comparable and robust genetic analysis. This detailed sampling strategy, along with the use of control samples from related shrimp species, enabled us to perform a robust genetic diversity analysis, ensuring the reliability and validity of our findings.

### 2.2. Total RNA Extraction and Purification

Instruments required for the experiment, such as forceps, scissors, and tissue-grinding rods, were soaked in 3% H_2_O_2_ overnight and then sterilized and dried at 121 °C for 20 min using a wet heat sterilization method. The muscle (MU), hepatopancreas (HE), lymphoid (LY), and intestinal tract (IN) tissues were collected from 3-month-old Pacific blue shrimp, placed in 1.5 mL microcentrifuge tubes containing 500 µL of RNA *later*^TM^ (#2724011, Invitrogen, Vilnius, Lithuania), and stored at −20 °C until use. The tissues were later removed, and the RNA *later*^TM^ was pressed out. Total RNA was extracted using the TRIzol^®^ (Invitrogen, Carlsbad, CA, USA) reagent. Tissue was weighed and homogenized in a microcentrifuge tube with a tissue-grinding rod, Bio Vortezer Mixer 1083-MC (BioSpec, Bartlesville, OK, USA), at a ratio of 100 mg/mL TRIzol^®^ (50 mg/mL for HE) and incubated for 5 min. To this, 0.2 mL/mL of TRIzol^®^ chloroform was added, shaken vigorously for 15 s, and incubated at room temperature for 3 min. The solution was then centrifuged at 12,000× *g*, 4 °C for 15 min, and the obtained supernatant was transferred to another microcentrifuge tube. To the solution, 0.5 mL of isopropanol (J.T Baker, Philipsburg, NJ, USA)/mL TRIzol^®^ was added, mixed well by shaking, and incubated at room temperature for 10 min. The solution was then placed in a freezer overnight at −20 °C and centrifuged at 12,000× *g*, 4 °C for 10 min to obtain a white precipitate. The supernatant was removed, and the precipitate was washed with 1 mL of 75% EtOH/mL TRIzol^®^, shaken, and centrifuged at 7500× *g*, 4 °C for 5 min. The supernatant was removed, and the precipitate was air-dried in a 55 °C incubator for 2–3 min. DEPC (diethyl pyrocarbonate)-treated 2dH_2_O was added to the precipitate, and the solution was heated at 55 °C for 15 min to dissolve the RNA. The total RNA concentration was measured using a NanoDrop^TM^ 1000 spectrophotometer (Thermo Fisher Scientific, Waltham, MA, USA) by measuring optical density OD_260_ and OD_280_. The RNA quality was checked by means of electrophoresis using fresh 1% agarose gel placed in the Muoid-2plus horizontal electrophoresis system (Advance, Tokyo, Japan), followed by staining with ethidium bromide for 10 min. DNA was removed using the DNase I, Amplification Grade (Invitrogen, Carlsbad, CA, USA). For DNase I digestion, 1 μg RNA, a 1 µL 10× DNase I Reaction Buffer, and 1 μL DNase I, AmpGrade (1 U/µL) were mixed with DEPC-treated 2dH_2_O to create a total volume of 10 μL, and the mixture was reacted at room temperature for 15 min on ice. Following that, 1 µL of 25 mM EDTA was added, and the solution was incubated at 65 °C for 10 min. The purified total RNA was stored at −80 °C for future use. Finally, the total RNAs obtained from the same organs of three individuals were pooled into separate tubes. Approximately 2 µg of total RNA was used to construct 14 high-quality cDNA libraries comprising 6 MU, 2 HE, 4 LY, and 2 IN samples. This pooling strategy was employed to reduce biological variability and achieve a more robust representation of the transcriptome for each tissue type. The different numbers of cDNA libraries for each tissue type were based on strategic decisions considering multiple factors, including the ease of obtaining high-quality RNA and the biological significance of each tissue type.

### 2.3. Establishment of Transcriptome Database and Annotation of Sequencing

To establish the transcriptome database, we employed the high-throughput sequencing platform Illumina HiSeq 4000 NGS system (Illumina, San Diego, CA, USA), which enabled the generation of large-scale transcriptome data. Clean reads were obtained after removing the reads with low quality, joint contamination, and a high N content of unknown bases in the original raw data. These were then used to calculate Q20, Q30, GC content, and sequence length. Subsequently, they were assembled de novo using Trinity V.2.0.6 [45]; noise from the assembled transcriptome was removed using the TGI Clustering tools (TGICL) [46], and the reads were sorted and screened to obtain unigenes. The obtained unigenes were divided into Clusters (representing a collection of unigenes with a similarity greater than 70%; the data names start with CL) and Singletons (representing individual unigenes; the data names start with unigenes). Finally, BLAST [47] was used to annotate the unigenes in the NT, NR, Clusters of Orthologous Groups (COG), Gene Ontology (GO), Kyoto Encyclopedia of Genes and Genomes (KEGG), and Swiss-Prot functional databases.

### 2.4. Genomic DNA Extraction

Approximately 0.06 ± 0.01 g of shrimp muscle tissue was placed in a 1.5 mL microcentrifuge tube, and 600 µL of 70% ethanol was added to it. The mixture was then stored at −20 °C until genomic DNA extraction. During DNA extraction, ethanol was discarded, and the tissue samples were washed twice with a 300 µL DE buffer (1 M Tris, 0.5 M EDTA, 10% SDS, and 2dH_2_O; pH 8). Afterward, a 600 µL DE buffer and 2 µL of proteinase K (Gene Mark, Taipei, Taiwan) were added to the tube, mixed evenly, and incubated in a rotary shaker (Evernew, Yu-Shing Biotech. Ltd., Taipei, Taiwan) at 55 °C and 80 rpm for 2–6 h until the tissue samples were completely dissolved. Following that, 500 µL of the sample was pipetted into a new 1.5 mL centrifuge tube, 300 µL of MPC Protein Precipitation Reagent (Epicentre Technologies Co., Madison, WI, USA) was added to it, and the contents were invert-mixed for approximately 2 min to precipitate the protein, and centrifuged at 10,000× *g* and 4 °C in a micro-refrigerated high-speed centrifuge for 10 min. Subsequently, 600 µL of the supernatant was transferred to a new 1.5 mL centrifuge tube, 600 µL of isopropanol (Sigma Chemical Co., St. Louis, MO, USA) was added to it, mixed for 20–40 min using a 3D rotary shaker (B3D1320, Benchmark Scientific, Sayreville, NJ, USA), and centrifuged at 4 °C and 10,000× *g* for 10 min in a micro-refrigerated high-speed centrifuge. After discarding the supernatant, 200 µL of 70% ethanol was added to wash the tube wall, and the mixture was allowed to stand for 10 min. After removing the alcohol, the pellet was washed using 200 µL of 70% ethanol and centrifuged at room temperature for 10 s using a mini again microcentrifuge (C1008, Benchmark Scientific, Sayreville, NJ, USA) to remove the alcohol in the tube completely before being oven-dried at a constant temperature of 55 °C. The pellet was then resuspended in 50 µL of 2dH_2_O and kept in a water bath maintained at 37 °C for 4–8 h until complete dissolution.

The concentration and absorbance of the DNA samples were measured using a NanoDrop^TM^ One spectrophotometer (Thermo Fisher Scientific, Waltham, MA, USA) by recording the OD_260_ and OD_280_ values. The DNA quality was assessed by means of electrophoresis separation on a 0.8% agarose gel and stained with a GelRed Nucleic Acid Gel Stain (Biotium Inc., Fremont, CA, USA) for 20 min. The results were visualized using the Slite140 Fluorescent Photoresist System (Avegene Life Sciences, Taipei, Taiwan). The DNA samples were diluted to a working concentration of 25 ng/µL and stored at −20 °C until further use.

### 2.5. Genetic Diversity Analysis

Microsatellite markers were searched in the Pacific blue shrimp transcriptome dataset and NCBI database [48,49,50], and a common microsatellite marker for Penaeus was designed using Geneious V.9.1.8 software (https://www.geneious.com accessed on 3 April 2022). The primers were designed using the following criteria: predicted product size, 100–500 bp; GC content, 40–60%; average *T*_m_, 52–62 °C; the self-adhesive area of the primer was avoided; and a self-adhesive temperature ≤37 °C. The 5′ ends of the forward primers of all microsatellite markers were connected with fluorescent primer adapters for the second PCR amplification. Specifically, markers focusing on the molecular markers involved in functional genes related to shrimp disease resistance traits were identified from the transcriptome dataset.

The first PCR amplification was carried out in a reaction volume of 10 µL comprising a 5 µL 2× *Taq* DNA polymerase Mix (Bioman, Taipei, Taiwan), 0.3 µL forward primer, 0.3 µL reverse primer, 2.4 µL 2dH_2_O, and 2 µL template DNA. PCR was performed using a 96-well polymerase chain reactor Veriti^®^ thermal cycler or SimpliAMP^TM^ thermal cycler (both from Applied Biosystems, Foster City, CA, USA) and the following reaction conditions: 94 °C for 5 min; 30 cycles of 94 °C for 40 s, *T*_m_ °C for 30 s, and 72 °C for 40 s; 72 °C for 8 min; and maintaining the temperature at 4 °C. The annealing temperatures (*T*_m_) varied according to the primer and ranged from 56 °C to 60 °C for the Pacific blue shrimp transcriptome dataset and from 52 °C to 60 °C for the NCBI database.

For the second PCR amplification, the original forward primer was replaced with a fluorescent primer tagged with FAM (blue), ROX (red), NED (yellow), or JOE (green) dyes. The reaction mix (10 µL) comprised the same components as those in the first PCR amplification, except for the replacement of the forward primer and template DNA with 0.3 µL of the fluorescent primer and 2 µL of the first PCR product diluted tenfold, respectively. PCR was performed using the same conditions and PCR systems as the first PCR amplification. Finally, the second PCR products were separated by means of electrophoresis on 2% agarose gel, stained with the GelRed^®^ Nucleic Acid Gel Stain for 25–30 min, and photographed using a Slite140 Compact Gel Documentation System (Avegene Life Sciences, Taipei, Taiwan).

Subsequently, 5 µL of each of the four fluorescently labeled PCR products were pooled with 0.2 µL GeneScan^TM^ 600 LIZ^TM^ size standard and 10.8 µL HiDi^TM^ deionized formamide (both from Applied Biosystems, Foster City, CA, USA) added to them, denatured at 94 °C for 3 min, and immediately transferred to ice for 5 min. The samples were then analyzed using an ABI PRISM^®^ 3730xl DNA Analyzer (Applied Biosystems, Foster City, CA, USA) to separate short tandem repeats (STRs). The data were analyzed and interpreted using Geneious V9.1.8 software, and the results were exported to Cervus 3.0.7 (http://www.fieldgenetics.com/pages/home.jsp accessed on 3 April 2022) to calculate the following genetic parameters: the mean number of alleles (*N*a), sum of all alleles in the population, average observed heterozygosity (*H*_o_), actual proportion of heterozygous individuals contained in each locus in the population; the average expected heterozygosity (*H*_e_); and estimated heterozygosity in the population according to the Hardy–Weinberg equilibrium [51]. The expected value of the degree of convergence was calculated along with two key parameters used to assess genetic diversity and population structure: polymorphic information content (PIC), which represents the amount of polymorphic information in linkage analysis [52], and the gene fixation index (*F*_IS_), which measures the degrees of inbreeding between individuals within the population. 

### 2.6. Development of Universal Microsatellite Markers

We designed 30 functional primer sets from our dataset, of which 15 primer sets were successfully amplified. In addition, we obtained 25 sets of primers from the Blue Shrimp Genetic Diversity Test panel that showed amplification under the set conditions of temperature testing. Finally, 40 primer pairs were evaluated to identify the universal microsatellite markers for shrimp.

The samples of *L. stylirostris*, *L. vannamei*, and *P. monodon* were categorized into the following five groups: (1) PBS-8, comprising a group of eight individuals including *L. stylirostris*; (2) PWS-1, comprising a single individual of *L. vannamei*; (3) PWS-8, comprising a group of eight individuals including *L. vannamei*; (4) BTS-1, comprising a single individual of *P. monodon*; and (5) BTS-8, comprising a group of eight individuals including *P. monodon*. Among these, eight shrimps of *L. vannamei* and *P. monodon* were sourced from three different origins. *L. stylirostris* was selected for the polymorphism analysis from genetically diverse species with different genotypes.

The PCR was performed in a reaction mixture of 10 µL comprising a 5 µL 2× *Taq* DNA polymerase Mix (Bioman, Taipei, Taiwan), 0.3 µL forward primer, 0.3 µL reverse primer, 2.4 µL 2dH_2_O, and 2 µL template DNA. The PCR conditions and amplification system were the same as those used for the first PCR amplification (described in Section 2.5).

The *T_m_* was adjusted according to the primers, and the exact length of the marker was confirmed using capillary electrophoresis. Five microliters of the reaction solution from each group were mixed in five tubes. Geneious V9.1.8 software was used to interpret and analyze the shared shrimp microsatellite markers for the microsatellite marker data. After the analysis of NGS data, amplified and diverse microsatellite markers were used for STR verification. The PCRs were performed in two steps: first and second PCR amplifications and the data were analyzed and interpreted as described in Section 2.5.

### 2.7. Statistical Analysis of Population Diversity

The obtained genetic data were imported into GenAlEx 6.502 [53] to statistically analyze the *N*a of each microsatellite locus, as well as various coefficients for evaluating the diversity of the population, including *H*_o_, *H*_e_, PIC, and *F*_IS_.

## 3. Results

### 3.1. High-Throughput NGS and Alignment of Reference Gene Sequences

Fourteen cDNA libraries were established from MU (*n* = 6), HE (*n* = 2), LY (*n* = 4), and IN (*n* = 2) tissues collected from *L. stylirostris* (*n* = 18). A total of 7.04 Gb of averaged data was produced, and 687.0366 M sequence reads were generated with a Q30 score of more than 92.98% (Appendix A). From these data, 53,263 unigenes were assembled, with a total length of 94,147,609 bp; an average length of 1767 bp; an N50 of 3478 bp; and GC content of 44.56% (Appendix A). As shown in Appendix A, the length of the reads varied from 300 to >3000 bp, with most unigenes (*n* = 9999) being 300 bp.

The functional annotation of the identified unigenes against publicly available databases is shown in Appendix A. A total of 24,066 unigenes were annotated in at least one functional database (Appendix A), wherein 3711 unigenes were co-annotated in all databases (Figure 1).

The Venn diagram analysis revealed that a total of 3711 unigenes were co-annotated in all databases (Figure 1). The BLASTx search against the NR protein database revealed substantial similarity between the unigene sequences and diverse organisms. The top matches included Nevada termite (*Zootermopsis nevadensis*; 12.99%), Daphnia pulex (*Daphnia pulex*; 6.92%), Atlantic horseshoe crab (*Limulus polyphemus*; 6.39%), Pacific white shrimp (*L. vannamei*; 3.69%), black tiger shrimp (*P. monodon*; 2.12%), and Pacific blue shrimp (*L. stylirostris*; 0.10%). The remaining matches accounted for 67.89% of the total hits (Appendix A). The BLASTn search of the unigenes sequences against the NT database identified 23,635 coding sequence (CDS) regions (*p* < 0.0001), while ESTScan predicted 7006 CDS regions (Appendix A).

### 3.2. COG, GO, and KEGG Annotation Results

The possible positions of the predicted CDS unigenes in the system classification were analyzed through the prediction and possible functional classification of the unigenes in the COG database. A total of 7978 unigenes, derived from combined transcriptome data of the muscle (MU), hepatopancreas (HE), lymphoid (LY), and intestine (IN) tissues, were subdivided into 24 functional categories in the COG database. The largest category annotated with the most unigenes (*n* = 2187) was “general function prediction only”, followed by “post-translational modification, protein turnover, and chaperones” (*n* = 1046) (Figure 2).

A total of 8828 unigenes were annotated in the GO database. In the biological process category, “cellular component organization and or biogenesis” was the most annotated term, with 4294 unigenes, followed by the “metabolic process”, with 3712 unigenes, and “single-organism process”, with 3027 unigenes. In the cellular component category, most unigenes were annotated to the term “cell” (*n* = 2978), followed by “cell parts” (*n* = 2953) and “membrane” (*n* = 2160). In the molecular function category, “binding” was the most annotated term with 4283 unigenes, followed by “catalytic activity” (*n* = 3954) and “transporter activity” (*n* = 525) (Figure 3).

Annotation of the unigenes with the KEGG database mapped 18,565 unigenes to 42 pathways, among which “signal transduction” was the most frequently annotated pathway, with 2981 unigenes, followed by “global and overview maps”, with 2587 unigenes, and “cancers: overview”, with 2336 unigenes. Other predominant pathways included “transport and catalyst” (*n* = 1805) and “cellular community” (*n* = 1700) (Figure 4).

### 3.3. Microsatellite DNA Marker Detection

The annotation of the unigenes using BLAST and ESTScan identified 30,641 CDSs. Among these, 36,415 microsatellites were detected in 18,657 unigenes using the MISA software (https://webblast.ipk-gatersleben.de/misa/ accessed on 8 June 2022) [54] based on the assembled unigene as a reference sequence. We identified 7400 single-, 17,735 double-, 8844 triple-, 1296 four-, 327 five-, and 813 six-base repeats (Figure 5, Table 1) through GO annotation, KEGG annotation, pathway enrichment analysis, locus detection software, function classification, microsatellite variation sequence, and other biological information analyses. Based on the predicted microsatellite position on the genomic DNA sequence (CDS), we identified 30 functional genes from the *L. stylirostris* transcriptome to develop candidate molecular markers (Table A1 and Appendix A).

### 3.4. Genetic Diversity Analysis of Pacific Blue Shrimp Broodstock

A total of 100 microsatellite markers, including 42 designed from transcriptomes (Type I microsatellite markers; Table A1 and Appendix A) and 58 obtained from previous studies on the diversity of *Litopenaeus populations* (Type II microsatellite markers; Appendix A), were used to study the genetic diversity of the cultured *L. stylirostris* species in Brunei. After the PCR temperature test, 22 sets (52.38%) of Type I markers and 18 sets (31.03%) of Type II markers were successfully amplified (Appendix A).

An analysis of the broodstock population of 200 *L. stylirostris* revealed that three DNA markers were polymorphic (CL1472.Contig9, LV13, and LV29), which could be stably amplified in the population of *L. stylirostris* and analyzed using STRs. Among the remaining 37 markers, CNM4 had a single-base slip, which was inconsistent with the predicted length of the microsatellite sequence, and the other 36 markers were monomorphic, exhibiting only one allele. The alleles detected by each marker were expressed in the order A, B, C, etc., according to the alleles of different fragment sizes and lengths. Among them, LV13 detected the most alleles (*n* = 4 alleles), wherein both LV13 and CL1472.Contig9 yielded four genotypes (Table 2).

Next, we used these three markers (CL1472.Contig9, LV13, and LV29) to assess genetic variation in the *L. stylirostris* population. The *H*_o_ values for these markers were 0.600, 0.505, and 0.513, respectively, with an average of 0.54 ± 0.05, while *H*_e_ values were 0.537, 0.375, and 0.390, respectively, with an average of 0.43 ± 0.09. The average PIC and *F*_IS_ values of these three markers were 0.41 ± 0.22 (0.333, 0.246, and 0.663, respectively) and 0.17 ± 0.27 (0.315, 0.339, and −0.135, respectively). The average PIC value indicated that this population was moderately polymorphic (0.25 < PIC < 0.5), whereas the positive average *F*_IS_ value suggested that the Pacific blue shrimp broodstock lacked genetic diversity and was in a state of inbreeding. The CL1472.Contig9 locus showed a negative *F*_IS_ value, indicating that this locus presents more heterozygotes. Therefore, it is necessary to supplement the stock with new seedlings to maintain the genetic polymorphism (Table 2).

### 3.5. Development of Universal Microsatellites for Shrimp

Forty markers were bonded to the DNA of the three shrimp species using PCR, and NGS was performed after grouping and mixing to analyze the bonding effects of the markers and the three shrimp species (Figure 6). In PBS-8, 37 markers were successfully amplified, but only 2 markers, CNM13 and LV13, showed diversity, while the rest amplified only one or two alleles. In PWS-1, 35 markers were successfully amplified, of which 3 were not specific enough; the amplified sequences were less and inconsistent, so they were not included in the follow-up STR verification analysis (Table A2). In PWS-8, among the 31 markers that were stably amplified, 21 showed gene diversity, and the remaining 10 resulted in only one or two genotypes. In BTS-1, 18 of the 28 successfully amplified markers showed insufficient specificity and less and inconsistent amplified sequences; therefore, they were excluded from the follow-up STR verification analysis. In BTS-8, 4 of the 10 stably amplified markers showed gene diversity, and the remaining 6 amplified only one or two alleles; these 4 markers also showed diversity in *L. vannamei*. Although there were still three markers that failed to amplify stably or showed poor adhesion efficiency in *L. stylirostris* species, the amplified detection ratio was still 92.5%. Upon calculating the expansion success rate of the other two penaeid shrimp species, the proportions tested for successful amplification in *L. vannamei* and *P. monodon* were 77.5% and 25%, respectively.

Based on the results of the shrimp and marker amplification mentioned above, 17 microsatellite markers were successfully amplified in Pacific blue and white shrimps. Pacific white shrimps exhibited diversity, with four microsatellite markers successfully associated with Pacific blue, Pacific white, and black tiger shrimps, indicating stable expansion and diversity. To confirm whether the above markers could be used to analyze related genetic parameters, using techniques such as STR genotyping and diversity analyses, each marker was verified via STR analysis in shrimps. Based on the above amplification results, the STRs were compared with those of the shrimps in PBS-8, PWS-1, PWS-8, BTS-1, and BTS-8 groups; these were then bonded with PCR fluorescent primers and sequenced.

Among the 17 markers stably amplified from the two types of shrimps, 7 markers (CNM1, CNM8, CNM12, CNM22, LV29, CL4833.Contig1, and Unigene6789) yielded consistent results with the expected fragment lengths. Although the remaining 10 markers had stable fluorescent signals, the length of the fluorescent signal fragments did not match the expected fragment length. A comparison of the fluorescence sequencing results of these 10 markers with the results of NGS revealed that 7 markers (CNM2, CNM3, CNM4, CNM13, CNM15, CL3247.Contig3, and Unigene7536) contained the originally predicted microsatellites. However, there were single-base variations at different positions within the microsatellite sequences, resulting in no diversity as initially anticipated. It also showed that the variation in another marker (CL5798) bonded to the STRs in Pacific white shrimp was a single-base variation, and the sequence did not contain the predicted microsatellites. A comparison of the bonding part of LV3 and the STRs in Pacific white shrimp with the NGS results identified other single-base microsatellite variations in the sequence in addition to the originally predicted diversity of microsatellites. Therefore, the primer used in this study was not suitable for the future diversity analysis of the target species. A comparison of the NGS results with the bonding part of CNM23 in the Pacific white shrimp genome revealed no variation in the originally predicted three-base repeat microsatellite. In contrast, diversity was observed in a two-base repeat microsatellite within the sequence, indicating its potential suitability for the diverse analysis of shrimp; however, the target microsatellite should be changed from TTA to GT repeats. The analysis revealed that the three prawns shared the microsatellite part for CL1472.Contig13, CL517.Contig2, Unigene5692, and Unigene7147. These four markers produced stable signals in STR fluorescence analysis; Unigene5692 yielded one genotype in the three prawns, CL517.Contig2 and Unigene7147 resulted in two or more genotypes in Pacific white and black tiger shrimps and CL1472.Contig9 identified eight genotypes in Pacific white shrimps (Table 3). These four markers can be used as common primers for the population diversity analyses of Pacific blue, Pacific white, and black tiger shrimps.

In addition to our findings on the genetic diversity of *Litopenaeus stylirostris*, we identified four microsatellite markers derived from the shrimp transcriptome database. These markers, namely CL1472.Contig9 [(GAG)_7_], CL517.Contig2 [(AAT)_8_], Unigene5692 [(GAG)_6_], and Unigene7147 [(TGA)_5_] exhibited cross-species amplification across *Litopenaeus stylirostris*, *Litopenaeus vannamei*, and *Penaeus monodon*, making them valuable tools for genetic studies in multiple shrimp species. Furthermore, the length and quantity of alleles at these microsatellite loci could be utilized to distinguish between different shrimp species. Importantly, our data revealed a significant reduction in genetic diversity in *Litopenaeus stylirostris*. Each of the four microsatellite loci corresponds to functional genes involved in crucial biological processes. CL1472.Contig9 [(GAG)_7_] corresponds to *myosin light chain kinase*, CL517.Contig2 [(AAT)_8_] to *caspase*, Unigene5692 [(GAG)_6_] to *smooth muscle-like transcript variant X3*, and Unigene7147 [(TGA)_5_] to *eukaryotic translation initiation factor 2* (Appendix A). Through bioinformatics analysis, we found that all four microsatellite sequences were located within the Open Reading Frame (ORF) of their respective genes. Interestingly, three of the microsatellite loci, CL1472.Contig9 [(GAG)_7_], Unigene5692 [(GAG)_6_], and Unigene7147 [(TGA)_5_] were situated within the Coding DNA Sequence (CDS). Furthermore, CpG islands were discovered within the CL1472.Contig9 [(GAG)_7_] and Unigene5692 [(GAG)_6_] loci, highlighting potential regulatory elements within these genomic regions. These findings underscore the importance of microsatellite markers in assessing genetic diversity and understanding functional gene variation in *L. stylirostris*. By elucidating the relationship between genetic markers and functional genes, our study contributed to the field of shrimp molecular genetics and offered insights into the genetic management of shrimp populations (Figure 7).

## 4. Discussion

*L. stylirostris* has great potential for the development of diverse species for farming in Asia [11]. However, the primary research areas concerning this species are its taxonomy [50,55] and disease [14,56,57], while research on the genetic management of germplasm conservation, selection, and breeding, as well as on the development of genetic polymorphism monitoring and measurement technology is limited [16,17]. To provide molecular tools for genetic studies or the identification of target aquatic species, it is essential to utilize reference marker information developed for a species over time [50] or cross-species marker sequences from closely related species. This should be followed by primer pair design and the testing of conditions [58]. However, in the case of non-mainstream species, the development of cross-species markers incurs high testing costs and is compounded by the limited availability of reference information.

In recent years, NGS has emerged as a cost-effective and efficient method for developing molecular markers in non-model species for which the genetic information available is limited and requires seed management [59,60,61,62]. The genome sequences of the members of Penaeidae, such as *L. vannamei* [63], *P. monodon* [64], and *Marsupenaeus japonicus* [65], were assembled using the Illumina platform. A transcriptome database can be used as a comparative reference for *L. stylirostris*, providing more genomic information for subsequent genetic breeding research and assisting in the development of the aquaculture industry and genetic management research on this species. This study used a high-throughput transcript sequencing platform that is widely applied to develop molecular markers. In addition to making significant progress in identifying population-scale and high-throughput functional markers, the study results have great research value.

In this study, the Illumina HiSeq NGS system was used for the first time to sequence, assemble, and annotate the transcriptomes of the MU, HE, LY, and IN tissues of the Pacific blue shrimp to establish a gene bank. The Q20 (97.40%), Q30 (93.66%), and GC (45.25–52.46%) values of each cDNA dataset were similar to those in the *L. vannamei* transcriptome database [66,67,68], indicating that the transcriptome was of good quality. In addition, in the distribution of species annotations in the NR database, because of the limited reference materials for *L. vannamei*, only 0.10% of the unigenes were annotated to *L. stylirostris*. In comparison, 3.69% and 2.12% of the unigenes were annotated to *L. vannamei* and *P. monodon*, respectively. The transcript sequencing data generated in this study can be used to analyze differentially expressed genes and identify key genes underlying important economic traits, such as those related to biotic and abiotic stresses. Previous studies have explored the potential of transcriptomics in identifying white spot syndrome virus in *M. rosenbergii* [69], neuronecrosis virus infection in seven-band grouper (*Hyporthodus septemfasciatus*) [70], growth-related traits in kuruma shrimp (*M. japonicus*) [71], and melanin deposition in the fillets of Atlantic salmon (*Salmo salar*) [72].

The annotation of the identified unigenes with COG, KEGG, and GO databases revealed the distribution of functional genes in *L. stylirostris*, identifying the potential functional genes and predicting their expression. The unigenes identified the “cellular process” (*n* = 4294), “cell” (*n* = 2978), and “binding” (*n* = 4283) terms in the biological processes, cellular components, and molecular function GO categories, respectively. KEGG database [73,74] annotation identified the enriched pathways in *L. stylirostris*. The generated data can be used to understand the interactions and functional effects of various molecular regulatory information transfers involved in the metabolism, genetic information processing, environmental information processing, various other cellular processes, and diseases of *L. stylirostris*. These data can also allow for follow-up research on gene function and metabolic pathways in cells and help discover the regulatory mechanisms of specific functional genes of *L. stylirostris*.

Among the total 53,263 unigenes in the Pacific blue shrimp transcriptomics dataset, 36,415 microsatellites were detected in 35.03% of unigenes (*n* = 18,657), which is lower than those detected in giant freshwater prawns (*M. rosenbergii*; 48.76%) [75] and banana shrimp (*Fenneropenaeus merguiensis*; 43.5%) [76], but higher than those detected in ridgetail white prawn (*Exopalaemon carinicauda*; 10.97%) [77]. However, this proportion of unigenes containing microsatellites in Pacific blue shrimp was similar to that in Pacific white shrimp (*L. vannamei*; 34.36%) [78] and freshwater ornamental shrimp (*Neocaridina denticulata*; 38.77%) [79]. On average, the microsatellites were 2.58 kbp distant, whichis a distance higher than the distances in *F. merguiensis* (1.02 kbp) and *M. rosenbergii* (0.93 kbp) but lower than those in *E. carinicauda* (6.6 kbp) and *N. denticulate* (4.95 kbp), and similar to those in *L. vannamei* (2.34 kbp). Among the mined SSRs, the proportion of mono- (7400, 20.32%), di- (17,735, 48.70%), and tri- (8844, 24.29%) nucleotides accounted for more than 93%. These observations are consistent with those of edible or ornamental crustacean species [75,76,77,78,79]. The two shrimp species *L. stylirostris* and *L. vannamei* showed a high degree of genetic similarity in the microsatellite structure analysis of transcripts.

The detection of polymorphisms in transcriptome-derived microsatellite markers can help us understand the genetic variation structure of functional genes and has several advantages, including high efficiency, strong transferability, and correlation with potential genes [80]. Some studies have indicated that the probability of polymorphism in transcriptome-derived microsatellites may not have a significant positive correlation with the strictness of evolutionary constraints [32,81]. However, when selecting microsatellites, the following recommendations are typically followed: (1) microsatellites that are composed of adenine (A) and thymine (T) bases, which may easily lead to sequence self-adhesion (hairpin) are avoided; (2) the primer length is optimized between 18 and 25 bp, and it is ensured that the annealing temperatures of all primers fall within ±1 °C; (3) the total length of the PCR product is ensured to be within the range of 100–300 bp, with GC content in the range of 40–60%; (4) primer–microsatellite repeat pairs with a distance of less than 20 bp are avoided [82]. In this experiment, the designs of the other primers were consistent with all the above recommendations, except that the total length of the PCR products in the case of all three sets of primers exceeded 300 bp. In addition, it has been pointed out that if cross-species amplification is to be performed, the PCR annealing temperature of DNA and primer bonding should first be adjusted for each species. The amplification primers from different species can be adjusted at different DNA bonding temperatures to identify the profile providing the clearest amplified products; the annealing temperature should be raised as much as possible to avoid non-specific amplification; most primers give the best results at 58 °C [83], which is consistent with the optimal temperature for most experimental designs. Although our study initially tested 100 sets of loci in *L. stylirostris*, only three markers proved effective for reliable genotyping. The low proportion of polymorphic loci may be attributed to limited genetic variation within the broodstock population or the specific genomic regions targeted by these markers. While the identified microsatellite markers provide valuable insights into blue shrimp genetic diversity and broodstock management, the full genetic variability within a population may not be captured using a limited number of loci. We acknowledge this limitation and recommend that future studies incorporate more microsatellite or single nucleotide polymorphism (SNP) markers and utilize advanced high-throughput sequencing and genotyping technologies for a comprehensive genetic assessment. By exploring a broader range of loci and including more diverse populations, future research can provide a more complete understanding of the genetic variability within and among blue shrimp populations.

In this study, among the 42 transcriptome-derived microsatellite marker primers, 22 successfully amplified the product; they exhibited a reliable peak signal pattern upon capillary electrophoresis, which was obtained through test amplification among cross-species samples of different Penaeidae members. Nine markers were identified as polymorphic, and 25 scorable fragments of 111–296 bp were obtained, with an average of 6.25 amplicons per primer and NA values ranging from 1 to 10 (mean = 3.5). Furthermore, the degree of polymorphism ranged from 50.0% to 100%, with an average of 81.52%, which is similar to that observed in previous studies investigating transcriptome-derived microsatellite markers in other crustaceans [32,79]. These nine markers could detect and present the most polymorphic information in *L. vannamei*, followed by *P. monodon* and *L. stylirostris*. Because samples of wild blue shrimp are not easy to obtain, white shrimp were used as a control sample with high diversity for comparison. Using 42 microsatellite markers developed from the blue shrimp transcriptome, it was observed that 9 (42.86%) loci retained diversity, although the diversity of blue shrimp was only 4.55%. In addition, 58 sets (Appendix A) of polymorphic primer pairs that have been used to construct the genetic linkage map were selected from the microsatellite marker library of *L. vannamei*. They were identified with 80% diversity in the same species, which is in agreement with previous analyses of the same species [49,50]. However, the diversity of blue shrimp was only 11.11%, and the overall polymorphism was reduced by approximately 50.83%. The analysis revealed that the microsatellite markers CL1472.Contig13, CL517.Contig2, Unigene5692, and Unigene7147 were successfully amplified and shared among the three prawn species (Pacific blue shrimp, Pacific white shrimp, and black tiger shrimp). While these markers have demonstrated utility in differentiating between species, their effectiveness for detailed population analysis within a single species may be limited. Future studies should incorporate additional microsatellite loci or SNPs to enhance the resolution of population genetic assessments. These findings underscore the necessity for the careful selection of genetic markers tailored to specific research objectives, whether it be species identification or population genetic diversity analysis.

Insufficient genetic management in aquaculture populations can undermine their resilience, rendering them more susceptible to diseases, environmental stress, and other adverse factors. Therefore, effective genetic management is crucial to safeguard the health and adaptability of aquaculture populations. When genes associated with immunity [84,85], reproduction [86,87], environmental stress response [68,88,89,90], behavior [91], and other functions experience a loss of diversity, the population’s capacity to adapt is compromised, thereby heightening the risk of encountering various ecological pressures. The analysis of functional markers corresponds to key functional genes associated with resistance traits against White Spot Syndrome Virus (WSSV) and Acute Hepatopancreatic Necrosis Disease (AHPND) found in the transcriptome sequencing dataset. These genes include *adenosine deaminase*, *c-type lectin* [92], *caspase*, *chitinase*, *eukaryotic translation initiation factor*, *myosin light chain*, *ras-like GTP-binding protein Rho*, relish, and *serine protease* (SP), which play crucial roles in disease resistance mechanisms. Numerous studies have underscored their critical role in disease resistance mechanisms [93,94,95,96,97,98]. However, it is evident that these disease resistance-related gene markers have experienced a reduction in genetic polymorphism within the blue shrimp broodstock, likely due to recent potential inbreeding depression affecting disease resistance capabilities. Furthermore, through detailed explanations of the development of microsatellite markers and their association with functional genes, this study systematically investigated the potential factors contributing to the decline in economic disease resistance traits in Pacific blue shrimp aquaculture due to prolonged inbreeding practices and a lack of effective genetic management. This comprehensive analysis not only elucidates the scientific significance of the developed microsatellite markers but also sheds light on the genetic management challenges faced by blue shrimp aquaculture. These findings have the potential to advance research in this field and provide valuable insights for developing strategies to mitigate the adverse effects of reduced genetic diversity and inbreeding depression in shrimp populations.

Subsequent research will investigate the impact of genetic diversity on the adaptive strategies of Pacific blue shrimp populations in response to environmental changes and new challenges. Furthermore, previous studies have documented the vertical transmission of potential microbial communities from shrimp eggs and nauplii derived from breeding to the juvenile stage [99]. Hence, in addition to the proper genetic management of aquaculture organisms, it is crucial to manage microbial communities in aquaculture pond water [100,101]. Therefore, the relationship between the genetic diversity of superior populations and their gut microbiome should be explored for more precise germplasm management.

## 5. Conclusions

This study aimed to develop transcriptome-derived cross-species microsatellite markers for genotyping and evaluating genetic polymorphisms in the Pacific blue shrimp species (*L. stylirostris*). Our findings demonstrate the successful development and utilization of these markers. The established transcriptome database offers and facilitates not only rapid primer selection for closely related species but also lays the groundwork for functional gene analysis and molecular marker development, particularly for economically important traits in selective breeding programs. Additionally, this study highlights the importance of establishing a suitable genetic management system to mitigate the risks of inbreeding depression and loss of genetic diversity when introducing new breeds in the future. Furthermore, the developed microsatellite markers play a crucial role in addressing key issues in blue shrimp aquaculture genetic management. By examining their association with functional genes, this study provides valuable insights into the underlying genetic mechanisms underlying disease resistance traits in aquaculture populations. These insights can inform future research directions and contribute to the development of effective strategies for maintaining the health and adaptability of aquaculture populations. The findings of this study can contribute to the genetic management analysis of the *L. stylirostris* population and enhance our understanding of the molecular genetics of this species. Moreover, the study aids in conserving and improving the shrimp species, ultimately facilitating the sustainable development of Pacific blue shrimps in aquaculture.

## Figures and Tables

**Figure 1 animals-14-01685-f001:**
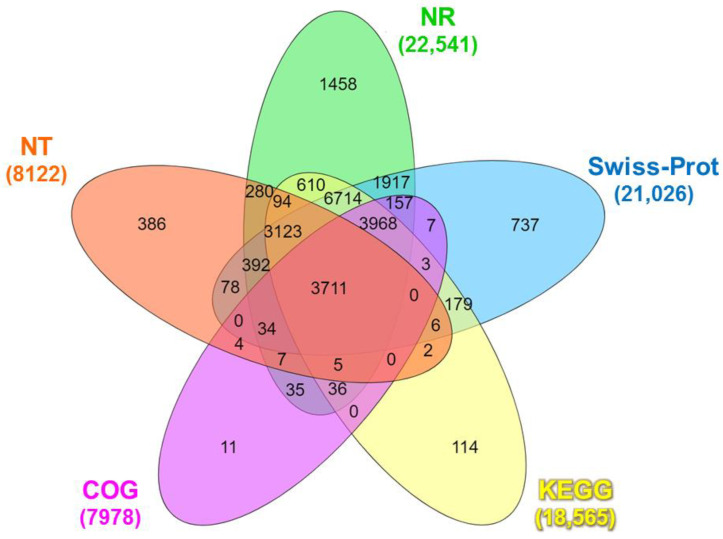
Venn diagram of the number of unigenes of *Litopenaeus stylirostris* functionally annotated in five major databases: NR, NT, COG, Swiss-Prot, and KEGG. NR: Unigenes with NCBI non-redundant protein; NT: nucleotide database; COG: Clusters of Orthologous Groups; Swiss-Prot: A curated protein sequence database that strives to provide high levels of annotation; KEGG: Kyoto Encyclopedia of Genes and Genomes.

**Figure 2 animals-14-01685-f002:**
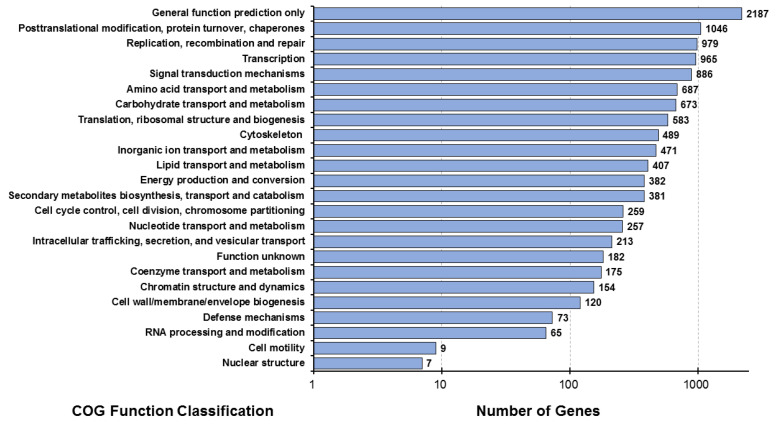
Distribution map of COG functional annotations of *Litopenaeus stylirostris*. The abscissa represents the number of unigenes corresponding to the classification, while the ordinate represents the COG functional classification. COG, Clusters of Orthologous Groups.

**Figure 3 animals-14-01685-f003:**
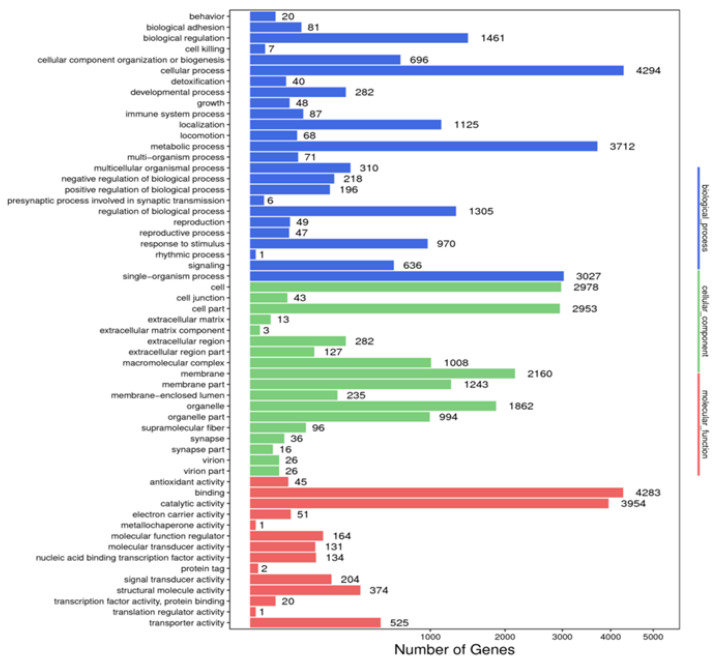
Annotated map of GO function of *Litopenaeus stylirostris*. The abscissa represents the number of unigenes in the corresponding category, while the ordinate represents the annotated terms in different GO categories. According to the NR annotation results, the unigene GO annotation information was obtained using BLAST2GO software (https://www.blast2go.com/, accessed on 28 May 2022). To preliminarily understand the gene function distribution of *L. stylirostris*, the sequenced unigenes, and assembled transcripts were compared with the GO database, and unigenes were mapped to different functional categories after predicting their possible functions. GO, Gene Ontology.

**Figure 4 animals-14-01685-f004:**
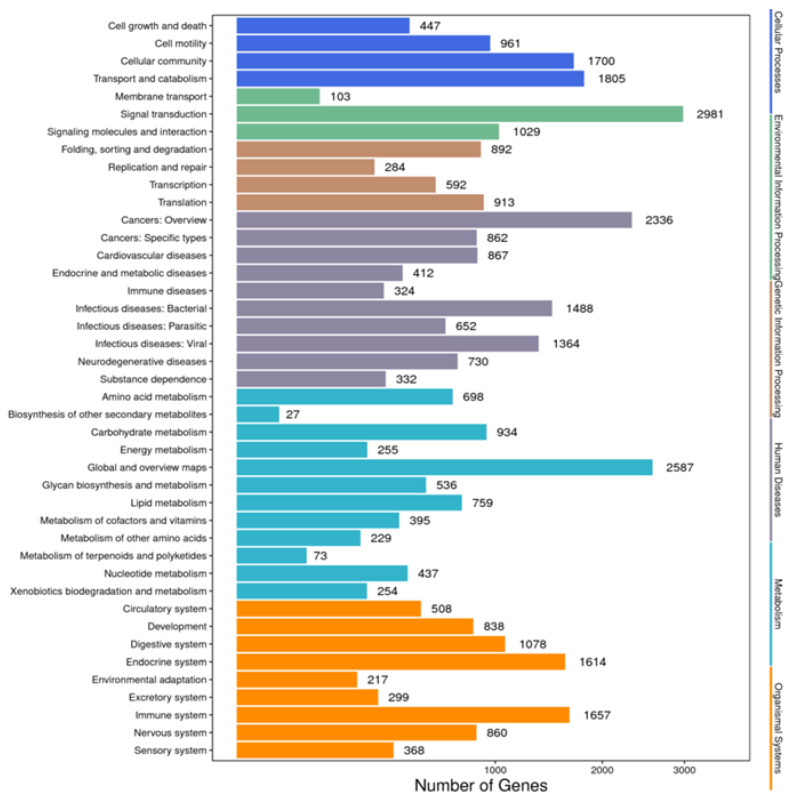
Functional annotations of the unigenes of *Litopenaeus stylirostris* using the KEGG pathway database. We predicted the potential functions of the unigenes in biochemical pathways and performed functional classification statistics to understand the overall distribution of gene functions in *L. stylirostris*. The x-axis represents the number of unigenes in each corresponding category, while the y-axis displays the functional subcategories of the KEGG pathway database. The six functional categories in the KEGG pathway database—cellular processes, environmental information processing, genetic information processing, human diseases, metabolism, and organismal systems—are represented on the right-hand y-axis. KEGG, Kyoto Encyclopedia of Genes and Genomes.

**Figure 5 animals-14-01685-f005:**
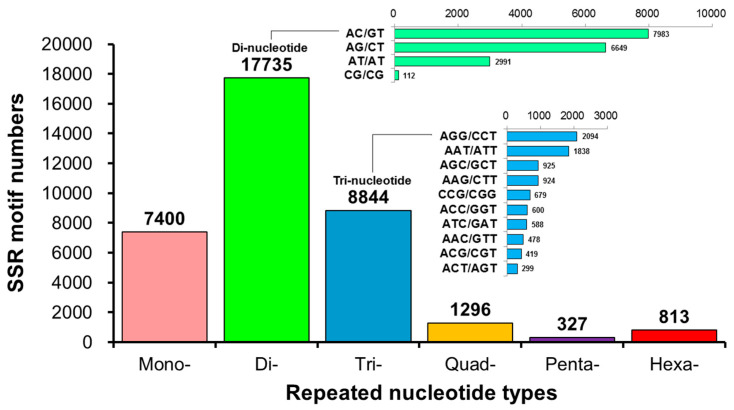
A statistical map of the distribution of microsatellite marker (or simple sequence repeat; SSR) length in *Litopenaeus stylirostris*. The horizontal axis displays the length of the repeat unit and specific repeat type, while the vertical axis represents the number of microsatellites. ‘Number of Repeats’ refers to the length of the repeated bases; ‘Mono-Nucleotide Repeats’, ‘Di-Nucleotide Repeats’, ‘Tri-Nucleotide Repeats’, ‘Quad-Nucleotide Repeats’, ‘Penta-Nucleotide Repeats’, and ‘Hexa-Nucleotide Repeats’ indicate the number of single-, double-, triple-, four-, five-, and six-base repeats, respectively.

**Figure 6 animals-14-01685-f006:**
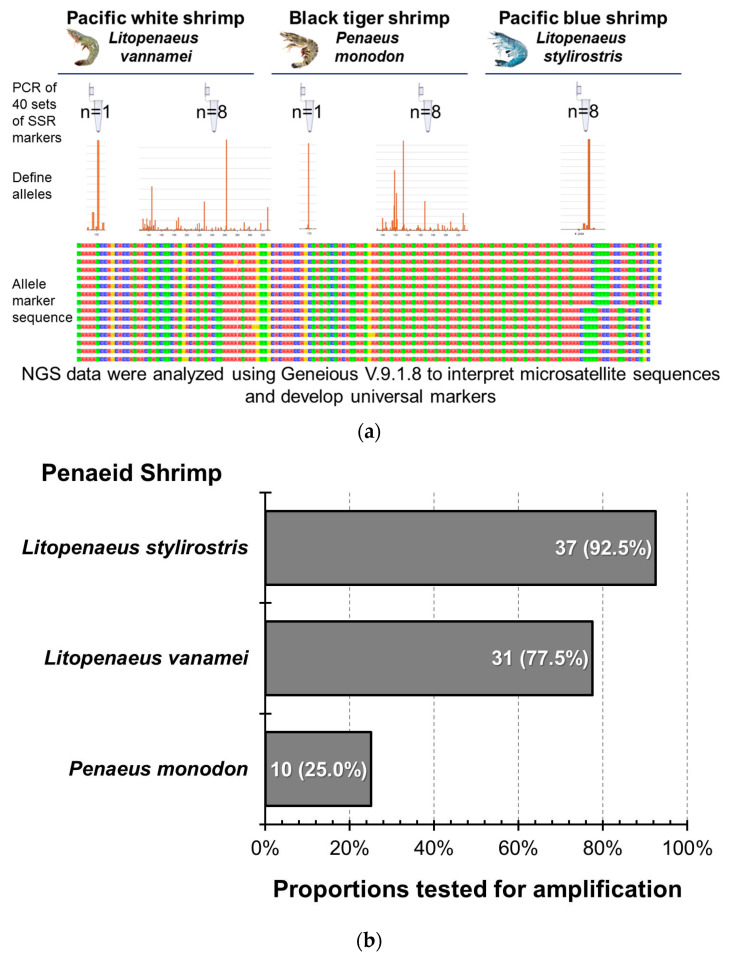
The amplification efficiency of microsatellite markers shared by different penaeid shrimp species. (**a**) Diagram of the total marker testing methodology using 40 microsatellite markers. (**b**) The successful amplification rate of the 40 markers in different Penaeid species. The colors in the allele marker sequence represent different bases: red for adenine (A), green for thymine (T), blue for cytosine (C), and yellow for guanine (G).

**Figure 7 animals-14-01685-f007:**
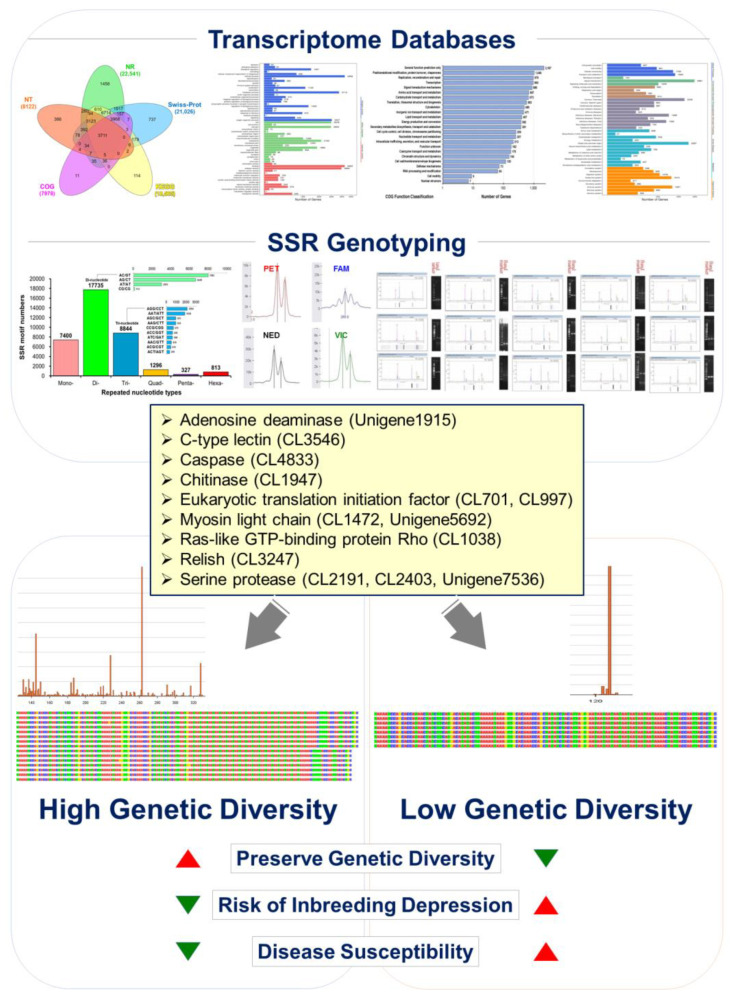
Illustrating genetic management strategies to mitigate the inbreeding decline in aquaculture blue shrimp (*Litopenaeus stylirostris*) populations. Drawing insights from the blue shrimp transcriptome gene database, we targeted functional genes associated with White Spot Syndrome Virus (WSSV) and Acute Hepatopancreatic Necrosis Disease (AHPND) resistance. These genes include *adenosine deaminase* (Unigene1915), *c-type lectin* (CL3546), *caspase* (CL4833), *chitinase* (CL1947), *eukaryotic translation initiation factor* (CL701, CL997), *myosin light chain* (CL1472, Unigene5692), *ras-like GTP-binding protein Rho* (CL1038), *relish* (CL3247), and *serine protease* (CL2191, CL2403, Unigene7536). By conducting the polymorphic analysis of each functional microsatellite DNA marker genotype, our study reveals that blue shrimp populations exhibit diminished genetic polymorphism due to prolonged inbreeding and insufficient genetic management. This decline in genetic diversity contributes to inbreeding depression, impacting economically vital breeding traits and disease resistance. Implementing effective genetic management strategies for genotyping can help alleviate inbreeding depression in aquaculture populations.

**Table 1 animals-14-01685-t001:** Statistics of the number of simple sequence repeat (SSR) marker types of transcripts in *Litopenaeus stylirostris*.

Number of Motifs	SSR Motif Unit					
	Mono-	Di-	Tri-	Quad-	Penta-	Hexa-
4–8	0	11,478	8327	1083	306	635
9–12	1764	3267	279	100	4	110
13–16	2938	478	93	54	10	44
17–20	936	260	65	32	5	18
21–24	767	320	51	23	2	6
25–28	315	446	12	2	0	0
>29	680	1486	17	2	0	0
Total	7400	17,735	8844	1296	327	813

**Table 2 animals-14-01685-t002:** Allele and genotype analysis of polymorphic microsatellite markers in *Litopenaeus stylirostris* populations ^1^.

Locus	Allele			Genotype		Polymorphism Statistics			
	Code	Length (bp)	*Freq.*	Codes	*Freq.*	PIC	*H* _o_	*H* _e_	*F* _IS_
CL1472. Contig9	A	111	0.495	AA	0.167	0.663	0.513	0.39	–0.135
	B	114	0.495	AB	0.656				
	C	117	0.010	BB	0.167				
				CC	0.01				
LV13	A	284	0.212	AA	0.193	0.333	0.6	0.537	0.315
	B	298	0.557	AB	0.17				
	C	318	0.080	BB	0.477				
	D	320	0.080	CD	0.159				
LV29	A	142	0.514	AA	0.386	0.246	0.503	0.375	0.339
	B	145	0.486	AB	0.257				
				BB	0.357				
				Mean		0.41 ± 0.22	0.54 ± 0.05	0.43 ± 0.09	0.17 ± 0.27

^1^ A total of 200 samples were analyzed from broodstock populations of *Litopenaeus stylirostris* (*n* = 200), including 100 male and 100 female shrimps.

**Table 3 animals-14-01685-t003:** Allelic statistics of microsatellite markers applied in penaeid shrimp.

Locus ^1^ [Motif]	*Litopenaeus stylirostris*		*Litopenaeus* *vannamei*		*Penaeus* *monodon*	
	Allele	Length (bp)	Allele	Length (bp)	Allele	Length (bp)
CL1472.Contig9 [(GAG)_7_]	A	166	A	175	A	158
	B	169	B	178		
	C	172	C	181		
			D	184		
			E	187		
			F	190		
			G	196		
			H	205		
CL517.Contig2 [(AAT)_8_]	A	111	A	115	A	112
	B	114	B	118	B	121
Unigene5692 [(GAG)_6_]	A	294	A	294	A	296
Unigene7147 [(TGA)_5_]	A	204	A	200	A	194
	B	207	B	206	B	200
			C	209	C	203

^1^ Microsatellite markers for all transcripts listed above are located in the CDS.

## Data Availability

The raw reads of the RNA sequencing data of *L. stylirostris* generated in this study have been submitted to the NCBI sequence read and deposited in the GenBank Data Archive under the BioProject accession number PRJNA957851.

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
