# Peer review of "Transcriptomic Insights and the Development of Microsatellite Markers to Assess Genetic Diversity in the Broodstock Management of Litopenaeus stylirostris"

_animals, 2024, doi:10.3390/ani14111685_

Round 1
Reviewer 1 Report
Comments and Suggestions for Authors
This manuscript provides theoretical guidance for the genetics and breeding of the Pacific blue shrimp by analyzing and exploring Pacific blue shrimp transcriptome and microsatellite markers. However, I think the manuscript needs to be improved before it could be considered for publication. The largest bottleneck is related to the poor description of the experimental method, which affects the understanding of the results.
The Material and Methods section needs to be revised to understand the validity of the results. The processes used to determine every parameter have to be clearly described to fully understand the obtained results. For example, reliability of transcriptomic data should be verified by RT-qPCR.
Line133-149: What are the different shrimp and tissues used for? All test samples should have a detailed description of their purpose.
Line135: Pacific blue shrimp were sourced from 2019, while Pacific white shrimp and Black tiger shrimp were sourced from 2021 as control samples. Why are the sampling times so different? When was the transcriptome sequencing performed and if the samples stored for a long period of time will affect the results of the experiment. In addition, the authors should add information on the individual size and growth stage of the shrimp.
Line142-Lin147: The authors should state the sex ratio of Pacific white shrimp and Black tiger shrimp, and describe how commercially available shrimp were assessed to meet the requirements of the experiment.
Line188: What kind of shrimp does the 14 high-quality cDNA libraries refer to? And please clarify whether the sample is a mixed one.
Line790: The discussion highlighted the critical role of the PCR annealing temperature of DNA in cross-species amplification. Consequently, it is recommended that the annealing temperature of the primers be included in the Table A1.
Line361: What data are COG, GO, and KEGG analyzed on? I don't quite understand if it's based on MU, HE, LY or IN.
Reviewer 2 Report
Comments and Suggestions for Authors
The author presents an interesting study development of microsatellite markers for broodstock management of Litopenaeus stylirostris. The bright side of the manuscript is that to provide important results on the current research on related topic. In this context, the study contributes to understanding the the genetic diversity and broodstock management of Litopenaeus stylirostris. However, some parts of the manuscript need to be improved. Therefore, I would like to make some suggestions to improve the quality of the paper as below:
Lines 17-29: The Simple Summary Section should be shortened. The main of the study, the method, the most important findings, and the main conclusion can be given with 6-7 sentences.
Lines 77-78: A reference is needed here.
Lines 104-106: “Microsatellite markers are simple sequence repeats of DNA motifs that are widely distributed throughout the genomes of eukaryotes. These are highly polymorphic, codominant, and have been widely used for exploring genetic diversity [27-29], population structure, parentage determination [30], and constructing genetic linkage maps [31-35].” The importance of Microsatellite markers should be emphasised with different studies with different species. Also references for different species should be added here. In this context, I think this sentence would better fit here -> “Microsatellite markers are simple sequence repeats of highly polymorphic and codominant DNA motifs that are widely distributed throughout the genomes of eukaryotes and are widely used for different population genetic studies in many different species such as genetic diversity [27-29], population structure, parentage determination [30, doi.org/10.3390/fishes8110544, doi.org/10.3390/biology12030401], constructing genetic linkage maps [31-35], animal identification and meat traceability (doi: https://doi.org/10.1016/j.foodcont.2017.03.017, doi.org/10.3390/genes13101825) and animal breeding (doi.org/10.3390/genes13010099, doi.org/10.3390/fishes6040047).”
Lines 125-129: “The findings of this study will contribute to the genetic management analysis of the L. stylirostris population and understanding the molecular genetics of this species. The study will also help conserve/improve the shrimp species and facilitate the sustainable development of Pacific blue shrimp aquaculture.” I think, these sentences would better fit to Conclusion Section.
Lines 708-724: The conclusion should be rephrased as follows; please start with a brief description of the study (the aim of the study with a sentence), explain the main findings of the study briefly (the results that the authors found), explain how your results contribute to to field with 2-3 sentences, and explain the limitations of the study and describe the future remarks briefly.
Author Response
請參閱附件。

Reviewer 3 Report
Comments and Suggestions for Authors
The study is innovative and interesting, but a major limitation of the whole study is that all the genetic diversity parameters were obtained for only three microsatellite loci. Two of them were inbred and one was outbred, so the results have to be interpreted with caution, which should be clearly emphasised in the discussion and conclusion sections of the paper, because usually several times more microsatellite loci are used in population genetics analysis.
Abstract: based on journal’ requirement it should be about 200 words, now it too large. Focus more on results and methodology, less on background and conclusions. L41-43 change whole names of indexes to symbols Ho, He, PIC, and FIS, they are widely used population genetic parameters. Also write what can be concluded from the results, moderate genetic variability and high level of inbreeding.
Introduction: If I understand correctly, your investigated species are also exploited for humans in natural environment not only in the farms, so have these populations experienced negative effects of selective harvesting which also affects the genetic parameters of these animals? I suspect selective harvesting on the size. L104 what is the size of repetitive motif? L125-129 these sentences are is more suited to a discussion and conclusions rather than an introduction, because it says how the results will be useful, at this stage we do not know what are the results.
Methods: When you list regents and instruments, please unify what you write in parentheses, only the producer/company or also the city and country. L251 please include ranges of Tm. L279-298 these formulas are well known, even bachelor students new these formulas, please discard them. [47] citing is out of topic. L328 I do not see full name of Ne coefficient above in methodology part.
Results: L433-437 I wanted to clarify, were the other markers monomorphic, i.e. having only one allele? Why do you think there is such a low proportion of polymorphic loci out of all the loci analysed? L452-453 you should highlight that one locus showed negative value. L522-523 the use of these markers is better suited to species separation than population analysis.
Discussion: the limitation of the current study needs to be discussed in detail
Round 2
Reviewer 3 Report
Comments and Suggestions for Authors
All my comments have been answered and the limitations of the study have been extensively discussed in the discussion, also mentioned in conclusions. I have no further comments, nor have I seen any substantive comments from the other reviewers, and I therefore propose to accept the manuscript